

# The density and biomass of mesozooplankton and ichthyoplankton in the Negro and the Amazon Rivers during the rainy season: the ecological importance of the confluence boundary

Ryota Nakajima[1], Elvis V. Rimachi[2], Edinaldo N. Santos-Silva[2],
Laura S.F. Calixto[2], Rosseval G. Leite[3], Adi Khen[1], Tetsuo Yamane[4],
Anthony I. Mazeroll[5,6], Jomber C. Inuma[7], Erika Y.K. Utumi[8] and Akira Tanaka[8]

[1] Scripps Institution of Oceanography, University of California San Diego, La Jolla, CA, USA
[2] Plankton Laboratory, Biodiversity Coordination, National Institute of Amazonian Research (INPA), Manaus, Amazonas, Brazil
[3] National Institute of Amazonian Research (INPA), Manaus, Amazonas, Brazil
[4] Biotechnology Laboratory, Amazonas State University (UEA), Manaus, Amazonas, Brazil
[5] Soka University of America, Aliso Viejo, CA, USA
[6] Amazon Research Center for Ornamental Fishes, Iquitos, Peru
[7] Centro de Projetos e Estudos Ambientais do Amazonas (CEPEAM), Manaus, Amazonas, Brazil
[8] Instituto Água Floresta e Vida, Manaus, Amazonas, Brazil

Corresponding author
Ryota Nakajima, rnakajima@ucsd.edu

## ABSTRACT

The boundary zone between two different hydrological regimes is often a biologically enriched environment with distinct planktonic communities. In the center of the Amazon River basin, muddy white water of the Amazon River meets with black water of the Negro River, creating a conspicuous visible boundary spanning over 10 km along the Amazon River. Here, we tested the hypothesis that the confluence boundary between the white and black water rivers concentrates prey and is used as a feeding habitat for consumers by investigating the density, biomass and distribution of mesozooplankton and ichthyoplankton communities across the two rivers during the rainy season. Our results show that mean mesozooplankton density ($2,730$ inds. m$^{-3}$) and biomass ($4.8$ mg m$^{-3}$) were higher in the black-water river compared to the white-water river ($959$ inds. m$^{-3}$; $2.4$ mg m$^{-3}$); however an exceptionally high mesozooplankton density was not observed in the confluence boundary. Nonetheless we found the highest density of ichthyoplankton in the confluence boundary ($9.7$ inds. m$^{-3}$), being up to 9-fold higher than in adjacent rivers. The confluence between white and black waters is sandwiched by both environments with low (white water) and high (black water) zooplankton concentrations and by both environments with low (white water) and high (black water) predation pressures for fish larvae, and may function as a boundary layer that offers benefits of both high prey concentrations and low predation risk. This forms a plausible explanation for the high density of ichthyoplankton in the confluence zone of black and white water rivers.

## INTRODUCTION

The region where two different hydrological regimes meet is characterized by strong physical and biological processes (*Walkusz et al., 2010*; *Bolotov, Tsvetkov & Krylov, 2012*). The boundary zone between two densities of waters is generally enriched in both dissolved and particulate organic matters as well as distinct planktonic communities as a result of their accumulation at this interface (*Hill & Wheeler, 2002*; *Walkusz et al., 2010*). Extensive research on oceanic fronts between coastal water and river plumes has shown that the boundary zone can lead to increased primary productivity (*Franks, 1992*), mechanically concentrating zooplankton (*Epstein & Beardsley, 2001*; *Morgan, De Robertis & Zabel, 2005*), and attracting tertiary consumers (*Grimes & Kingsford, 1996*). Thus, the boundary zone is important for local ecosystem functioning.

The Amazon River is well-known for its largest and most dense river network in the world and has the highest level of discharge, contributing with ca. 20% to the total global continental water discharge into the oceans (*Sioli, 1984*). In the center of the Amazon basin, muddy white water of the Amazon River (locally named Rio Solimões) meets with black water of the Negro River, one of the largest tributaries, creating a conspicuous visible boundary spanning over 10 km along the Amazon River (Fig. 1). The black water of the Negro River is derived from the high concentration of humic substances, while the white water of the Amazon River is derived from highly suspended inorganic materials (*Sioli, 1984*; *Furch & Junk, 1997*; *Junk et al., 2015*). The water properties of the white and black waters are different in terms of many parameters such as flow speed, conductivity, turbidity, pH, water temperature, nutrient concentrations, and dissolved and particulate organic matter concentrations (*Laraque et al., 1999*; *Moreira-Turcq et al., 2003*; *Leite, Silva & Freitas, 2006*; *Filizola et al., 2009*; *Laraque, Guyot & Filizola, 2009*; *Franzinelli, 2011*; *Röpke et al., 2016*). Due to these differences, the black and white water rivers are not completely mixed until over 100 km beyond the confluence (*Laraque, Guyot & Filizola, 2009*).

Zooplankton are one of the central players in the Amazon River ecosystem, acting as a trophic link between primary producers and higher trophic levels including planktivorous fish (*Araujo-Lima et al., 1986*; *Hawlitschek, Yamamoto & Neto, 2013*). The conspicuous boundary between black and white water rivers may be ecologically important as it may act as a mechanical aggregator of zooplankton, and contribute to the subsequent attraction of consumers such as fish larvae. However, the density and biomass of zooplankton at the confluence remains unclear from a quantitative perspective. To date, most studies on zooplankton in this region have been conducted in the floodplain lakes associated with large black and white water rivers (*Brandorff, 1978*; *Robertson & Hardy, 1984*; *Trevisan & Forsberg, 2007*; *Ghidini & Santos-Silva, 2011*), but studies from large rivers are scarce (*Robertson & Hardy, 1984*) and there is no comparison of zooplankton between black and white water rivers. Similarly, previous studies investigated zooplankton in the floodplain lakes of mixed waters from black and white water rivers (*Trevisan & Forsberg, 2007*; *Caraballo, Forsberg & Leite, 2016*), yet very little is known about the boundary interface between white and black water rivers.

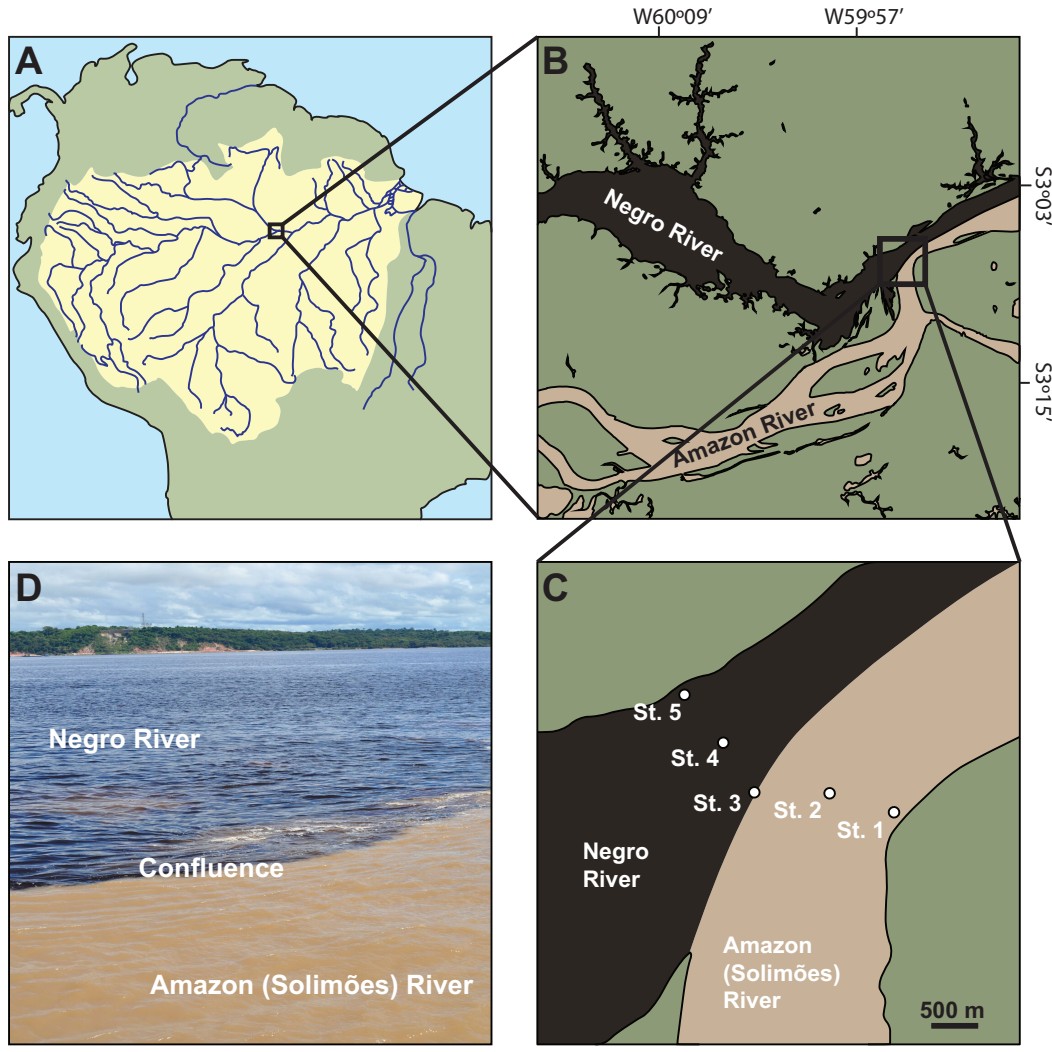

**Figure 1** **Location of the study sites.** (A) the Amazon Basin in South America. (B) the Amazon River (locally named Rio Solimões) and the Negro River in the center of the Amazon basin. (C) sampling sites across the two rivers: bank (St. 1) and center (St. 2) of the Amazon River, the confluence (St. 3), and center (St. 4) and bank (St. 5) of the Negro River. (D) the confluence.

Here, we tested the hypothesis that the confluence boundary between the white water of the Amazon River and the black water of the Negro River concentrates potential planktonic prey for consumers. For this purpose we examined (1) the density, biomass and composition of mesozooplankton in black and white water rivers and (2) the density and composition of mesozooplankton and ichthyoplankton at the confluence and how much they differ in respect to the black and white water rivers.

## MATERIALS & METHODS
### Study sites
The study was conducted in the center of the Amazon basin where the white water of the Amazon River (locally named Rio Solimões) and the black water of the Negro River (locally

named Rio Negro) merge in Manaus, Brazil (Fig. 1). All experiments and preparation of samples were carried out using the facilities of Centro de Projetos e Estudos Ambientais do Amazonas (CEPEAM) on the banks of the Negro River. The sampling of mesozooplankton was conducted at five sites across the rivers: the bank (St. 1) (S03°07′36.35″;W59°53′10.25″) and center (St. 2) (S03° 07′29.89″;W59°53′30.92″) of the Amazon River, the confluence (St. 3) (S03°07′29.64″;W59°53′55.10″), and the center (St. 4) (S03°07′13.43″;W59°54′05.19″) and bank (St. 5) (S03°06′57.97″;W59°54′17.74″) of the Negro River (Fig. 1). The bottom of the Amazon River is characterized by muddy and sandy sediments, while the river bottom of the Negro River by hard bedrocks (*Junk et al., 2015*). The water depths at the five sites were 11 m (St. 1), 72 m (St. 2), 44 m (St. 3), 62 m (St. 4) and 6 m (St. 5), which were measured by a measuring rope with a 20 kg weight.

## Sample collection

We collected mesozooplankton (including ichthyoplankton) from March 8–12, 2012 during the rising water period (rainy season). In total, six samplings were conducted at each sampling site. Mesozooplankton and ichthyoplankton were sampled by pooling three vertical tows of a plankton net (mesh size, 180-μm; diameter, 30 cm; length, 100 cm) equipped with a flowmeter (Rigo) from 10 m depth to the surface, except at St. 5 where towing was done from 5 m depth. Due to a large amount of sand and detrital particles such as plant debris, especially in the white water, the net was washed after every towing in order to reduce net clogging. The pooled samples were immediately brought back to the field laboratory within 30 min, and fixed with buffered formalin to a final concentration of 5% for subsequent microscopic observation.

Prior to the plankton collection, transparency was measured using a Secchi disc and water temperature was measured with a mercury thermometer at each site. In addition, surface water was sampled by a 10 L bucket at three sites (St. 1, 3 and 5) for the analyses of chlorophyll-*a* (chl-*a*), particulate organic carbon (POC) and nitrogen (PON) concentrations. The collected water (10 L) from each of the three sites was pre-filtered through a 180-μm mesh screen to remove zooplankton and the water samples were brought back to the laboratory along with the plankton samples.

## Sample analysis

For chlorophyll analysis, duplicate subsamples (50–100 mL each from bucket) were filtered onto GF/F filters (25 mm; Whatman, Little Chalfont, Buckinghamshire, UK), then immersed in 90% acetone and stored at 5 °C for 24 h. After centrifugation at 3,000 rpm for 5 min, the concentrations of chl-*a* were determined using a spectrometer (UV mini 1240; Shimadzu, Kyoto, Japan) according to the equation of *Ritchie (2006)*. For POC and PON analysis, duplicate subsamples (100–200 mL from bucket) were filtered onto pre-combusted (500 °C, 4 h) GF/F filters (25 mm, Whatman), and then dried for 24 h at 60 °C and stored in a desiccator until analysis. The concentration of POC and PON was measured using a CN analyzer (Fisons EA 1108 CHNS/O).

Mesozooplankton were identified to the lowest possible taxonomic level and counted under a dissecting microscope (Leica MZ9.5). Upon observation, large debris (e.g., wood

**Table 1** **Length-weight regression equations used for biomass calculations of different mesozoo-plankton taxa.**

| Taxonomic group | Equation | Source |
|---|---|---|
| Cladocerans | | |
| *Bosmina* sp. | ln DW ($\mu$g) $= 2.68$ ln L (mm) $+ 2.479$ | *Maia-Barbosa & Bozelli (2005)* |
| *Bosminopsis* sp. | ln DW ($\mu$g) $= 2.221$ ln L (mm) $+ 1.808$ | *Maia-Barbosa & Bozelli (2005)* |
| *Ceriodaphnia cornuta* | ln DW ($\mu$g) $= 1.888$ ln L (mm) $+ 1.442$ | *Maia-Barbosa & Bozelli (2005)* |
| *Chydorus* sp. | ln DW ($\mu$g) $= 3.93$ ln L (mm) $+ 4.493$ | *Dumont, Van de Velde & Dumont (1975)* |
| *Daphnia gessneri* | ln DW ($\mu$g) $= 3.22$ ln L (mm) $+ 1.169$ | *Azevedo et al. (2012)* |
| *Diaphanosomoa birgei* | ln DW ($\mu$g) $= 1.738$ ln L (mm) $+ 1.653$ | *Maia-Barbosa & Bozelli (2005)* |
| *Diaphanosoma* sp. | ln DW ($\mu$g) $= 2.22$ ln L (mm) $+ 1.140$ | *Azevedo et al. (2012)* |
| *Macrothrix* sp. | ln DW ($\mu$g) $= 3.177$ ln L (mm) $+ 2.850$ | *Azevedo et al. (2012)* |
| *Moina* sp. | ln DW ($\mu$g) $= 1.549$ ln L (mm) $+ 0.149$ | *Maia-Barbosa & Bozelli (2005)* |
| Other cladocerans | ln DW ($\mu$g) $= 2.653$ ln L (mm) $+ 1.751$ | *Bottrell et al. (1976)* |
| Copepods | | |
| *Argyrodiaptomus* sp. | ln DW ($\mu$g) $= 2.560$ ln L (mm) $+ 2.440$ | *Azevedo et al. (2012)* |
| *Notodiaptomus* sp. | ln DW ($\mu$g) $= 2.160$ ln L (mm) $+ 2.290$ | *Azevedo et al. (2012)* |
| Other calanoids | ln DW ($\mu$g) $= 3.150$ ln L (mm) $+ 2.470$ | *Azevedo et al. (2012)* |
| *Eucyclops* sp. | ln DW ($\mu$g) $= 2.40$ ln L (mm) $+ 1.953$ | *Bottrell et al. (1976)* |
| *Mesocyclops* sp. | ln DW ($\mu$g) $= 2.556$ ln L (mm) $+ 1.211$ | *Shumka, Grazhdani & Nikleka (2008)* |
| *Thermocyclops decipiens* | ln DW ($\mu$g) $= 3.244$ ln L (mm) $+ 1.570$ | *Azevedo et al. (2012)* |
| *Thermocyclops minutus* | ln DW ($\mu$g) $= 2.770$ ln L (mm) $+ 1.340$ | *Azevedo et al. (2012)* |
| Other cyclopoids | ln DW ($\mu$g) $= 2.40$ ln L (mm) $+ 1.953$ | *Bottrell et al. (1976)* |
| All nauplii | ln DW ($\mu$g) $= 2.40$ ln L (mm) $+ 1.953$ | *Bottrell et al. (1976)* |
| Insect larvae | | |
| Chaoboridae (diptera) | ln DW (mg) $= 2.692$ ln L (mm) $- 5.992$ | *Benke et al. (1999)* |
| Tipulidae (diptera) | ln DW (mg) $= 2.681$ ln L (mm) $- 5.843$ | *Benke et al. (1999)* |
| Chironomidae (diptera) | ln DW (mg) $= 2.618$ ln L (mm) $- 6.320$ | *Benke et al. (1999)* |
| Other diptera | ln DW (mg) $= 2.692$ ln L (mm) $- 5.992$ | *Benke et al. (1999)* |
| Coleoptera | ln DW (mg) $= 2.910$ ln L (mm) $- 4.867$ | *Benke et al. (1999)* |

**Notes.**
DW, dry weight; L, body length; ln, natural logarithm ($\log_e$).

and plant debris) was removed from the samples as much as possible, and then rose bengal was added to facilitate the separation of organisms from suspended matter. Large zooplankton and/or rare species (e.g., larval insects and calanoid copepods) and ichthyoplankton were first counted and sorted out, then the remaining was split (1/2–1/16), from which all zooplankton were characterized and enumerated. At least 300 zooplankton were enumerated in each sample. Copepods and cladocerans were identified to species level and insect larvae and ichthyoplankton to family level whenever possible.

The body length of copepods, cladocerans and insect larvae was measured using an eyepiece micrometer. The length measurements of zooplankton individuals were converted to dry weight (*DW*, mg) using previously reported length-weight regression equations (Table 1). The biomass (*B*, mg m$^{-3}$) of a given taxonomic group was estimated

**Table 2 Hydrological data.** Average (mean ± SD) water temperature (WT), transparency (Secchi depth), chlorophyll-*a* (chl-*a*), particulate organic carbon (POC) and nitrogen (PON) in the Amazon River (St. 1-2), the confluence (St. 3), and the Negro River (St. 4-5) in the center of the Amazon basin. *P* values indicate the differences in the values between the Amazon Rivers and the Negro Rivers, tested by Student's *t*-test. *P* values for WT and transparency were from the comparison between the average of St. 1-2 and St. 4-5, while those for Chl-*a*, POC, PON and C/N were derived from the comparison between St. 1 and 5. ND, no data.

| | Amazon River | | Confluence | Negro River | | *P* |
| --- | --- | --- | --- | --- | --- | --- |
| | **Bank** | **Center** | | **Center** | **Bank** | **(Amazon vs Negro)** |
| | (St. 1) | (St. 2) | (St. 3) | (St. 4) | (St. 5) | |
| WT (°C) | 27.5 ± 0.3 | 27.5 ± 0.2 | 27.6 ± 0.2 | 28.2 ± 0.9 | 28.0 ± 0.8 | 0.022 |
| Secchi depth (m) | 0.28 ± 0.06 | 0.29 ± 0.02 | 0.36 ± 0.03 | 1.14 ± 0.12 | 1.17 ± 0.15 | <0.001 |
| Chl-*a* (µg L$^{-1}$) | 3.77 ± 0.32 | ND | 2.10 ± 0.15 | ND | 1.97 ± 0.21 | 0.034 |
| POC (µg L$^{-1}$) | 1,262 ± 420 | ND | 881 ± 144 | ND | 446 ± 62 | 0.029 |
| PON (µg L$^{-1}$) | 333 ± 23 | ND | 316 ± 27 | ND | 114 ± 3 | <0.001 |
| C/N | 3.8 ± 1.1 | ND | 2.8 ± 0.3 | ND | 3.9 ± 0.6 | 0.86 |

based on its density ($A$, inds. m$^{-3}$) and individual dry weight: $B = A \times DW$. Reported length-weight regressions of some species that occur at the sampling site were not available, but we used regressions according to similar genera or shapes. Regressions established in tropical waters were also used when possible.

## Statistical analysis

The difference in the environmental factors and density and biomass of mesozooplankton and ichthyoplankton between the Amazon River (mean of St. 1-2) and the Negro River (mean of St. 4-5) was determined using two-sided Student's *t*-test. The difference in the density of ichthyoplankton between different sites was determined using one-way ANOVA and then differences among means were analyzed using Tukey-Kramer multiple comparison tests. A difference at $P < 0.05$ was considered significant.

Spatial similarities of mesozooplankton assemblage structure were graphically depicted using non-metric multidimensional scaling (MDS) and group average clustering was carried out. The similarity matrix obtained from the density values was calculated by the Bray-Curtis index (*Bray & Curtis, 1957*) with square-root transformed data. To test for spatial variation in community density, analysis of similarities (ANOSIM) was then undertaken (*Clarke & Warwick, 1994*). All multivariate analyses were conducted with the software PRIMER v. 6 (Plymouth Marine Laboratory).

## RESULTS

### Environmental factors and the structure of the confluence

Water temperature, transparency, and the concentration of chl-*a*, POC and PON were consistently distinct for white and black water rivers (Table 2). The values in the confluence in general were in the middle between black and white water rivers. The average (mean ± SD) surface water temperature in black water (28.1 ± 0.1 °C, mean of St. 4-5) was significantly higher by 0.61 ± 0.59 °C than that in white water (27.5 ± 0.8 °C, mean of
**Table 3  Spatial variation in the density of mesozooplankton.** Average (mean $\pm$ SD) density (ind. m$^{-3}$) of cladocerans, copepods, insect larvae and total mesozooplankton in the Amazon River (St. 1-2), the confluence (St. 3), and the Negro River (St. 4-5) in the center of the Amazon basin. $P$ values indicate the differences in the values between the Amazon Rivers (average of St. 1-2) and the Negro Rivers (average of St. 4-5) tested by Student's $t$-test.

| | Amazon River (white water) | | Confluence | Negro River (black water) | | $P$ |
| | Bank | Center | | Center | Bank | (Amazon vs Negro) |
| | (St. 1) | (St. 2) | (St. 3) | (St. 4) | (St. 5) | |
|---|---|---|---|---|---|---|
| Cladocerans | $834 \pm 250$ | $414 \pm 266$ | $1,363 \pm 635$ | $1,924 \pm 886$ | $1,999 \pm 947$ | <0.001 |
| Copepods | $453 \pm 107$ | $210 \pm 112$ | $860 \pm 797$ | $1,047 \pm 629$ | $479 \pm 166$ | 0.013 |
| Insect larvae | $4.2 \pm 3.2$ | $3.4 \pm 1.2$ | $4.9 \pm 1.3$ | $4.1 \pm 2.5$ | $6.0 \pm 3.6$ | 0.28 |
| Total | $1,291 \pm 271$ | $627 \pm 366$ | $2,228 \pm 1,388$ | $2,975 \pm 1,232$ | $2,484 \pm 1,068$ | <0.001 |

St. 1-2). Transparency (Secchi depth) was significantly lower in white water ($0.28 \pm 0.04$ m, mean of St. 1-2) than black water ($1.16 \pm 0.12$ m, mean of St. 4-5) (Table 2).

The chl-$a$ concentrations were significantly higher in white water, being 1.9-fold higher than in black water (Table 2). POC and PON concentrations in white and black rivers also significantly differed, being 2.8–2.9-folds higher in black water river (Table 2). C/N ratio was comparable between black and white water rivers (3.8–3.9), but lower in the confluence (2.8)

## Mesozooplankton density and biomass

The highest density ($2,817 \pm 1,162$ inds. m$^{-3}$, mean $\pm$ SD) and biomass ($5.14 \pm 2.55$ mg m$^{-3}$) of mesozooplankton were observed at the center of the Negro River (St. 4), while the lowest density ($577 \pm 345$ inds. m$^{-3}$) and biomass ($1.30 \pm 0.46$ mg m$^{-3}$) were observed at the center (St. 2) of the Amazon River (Figs. 2A and 2B). The mesozooplankton density and biomass in the black water river ($2,730 \pm 1,129$ inds. m$^{-3}$; $4.82 \pm 2.22$ mg m$^{-3}$, mean of St. 4-5) significantly exceeded those of white water river ($959 \pm 463$ inds. m$^{-3}$; $2.36 \pm 1.33$ mg m$^{-3}$, mean of St. 1-2) by 2.8 and 2.0-fold higher, respectively (Table 3). At the confluence (St. 3), the mesozooplankton density ($2,060 \pm 1,269$ inds. m$^{-3}$) showed intermediate values between black and white water rivers, while the biomass ($4.70 \pm 3.28$ mg C m$^{-3}$) was comparable to that in the black water river. The mesozooplankton density and biomass in the confluence showed the highest value among the sampling sites two times out of a total of 6 sampling times.

Cladocerans were the most dominant group in terms of density, contributing with 66.2%–82.2% to the total mesozooplankton density at all sites, followed by copepods (19.7–41.7%) and insect larvae (0.1–0.6%) (Fig. 2A). On the contrary, copepods were the most important in terms of biomass, contributing with 64.0–79.1% to the total mesozooplankton biomass, followed by cladocerans (13.4–20.9%) and insect larvae (6.5–17.4%) (Fig. 2B).

The density and biomass of cladocerans in the black water river were significantly higher ($1,962 \pm 875$ inds. m$^{-3}$; $0.92 \pm 0.42$ mg m$^{-3}$, mean of St. 4-5) than those of the white water river ($621 \pm 330$ inds. m$^{-3}$; $0.37 \pm 0.19$ mg m$^{-3}$, mean of St. 1-2) (Tables 3 and 4). In total, 26 species of cladocerans were observed (Table S1), among which *Diaphanosoma polyspina* was the most dominant taxa at all sites, contributing with 33.4%-65.5% to the

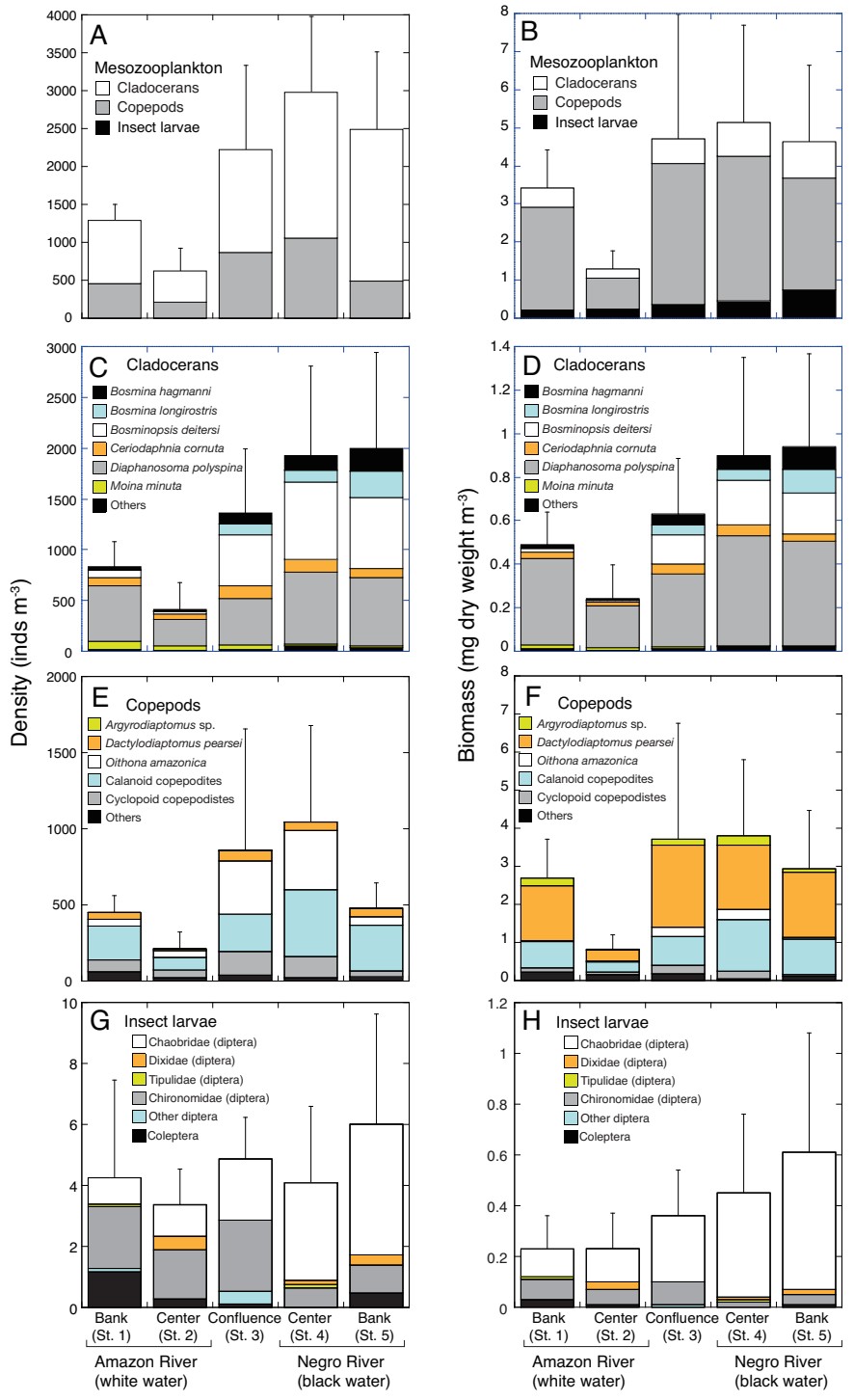

**Figure 2  Spatial variations in density and biomass of mesozooplankton.** Average (mean ± SD) density and biomass of (A, B) total mesozooplankton, (C, D) cladocerans, (E, F) copepods, and (G, H) insect larvae in the Amazon River (St. 1-2), the confluence (St. 3), and the Negro River (St. 4-5) in the center of the Amazon basin. Error bars represent standard deviation (SD) of abundance or biomass for six replicate measurements. Each legend category indicates the proportion of each taxon per mean.

**Table 4 Spatial variation in the biomass of mesozooplankton.** Average (mean ± SD) biomass (mg dry weight m$^{-3}$) of cladocerans, copepods, insect larvae and total mesozooplankton in the Amazon River (St. 1-2), the confluence (St. 3), and the Negro River (St. 4-5) in the center of the Amazon basin. $P$ values indicate the differences in the values between the Amazon Rivers (average of St. 1-2) and the Negro Rivers (average of St. 4-5) tested by Student's $t$-test.

| | Amazon River (white water) | | Confluence | Negro River (black water) | | $P$ |
| | Bank | Center | | Center | Bank | (Amazon vs Negro) |
| | (St. 1) | (St. 2) | (St. 3) | (St. 4) | (St. 5) | |
|---|---|---|---|---|---|---|
| Cladocerans | 0.49 ± 0.15 | 0.24 ± 0.16 | 0.63 ± 0.25 | 0.90 ± 0.45 | 0.94 ± 0.43 | <0.001 |
| Copepods | 2.70 ± 1.02 | 0.83 ± 0.36 | 3.71 ± 3.05 | 3.79 ± 2.00 | 2.95 ± 1.53 | 0.016 |
| Insect larvae | 0.22 ± 0.14 | 0.23 ± 0.14 | 0.36 ± 0.18 | 0.45 ± 0.31 | 0.61 ± 0.47 | 0.018 |
| Total | 3.41 ± 1.01 | 1.30 ± 0.46 | 4.70 ± 3.28 | 5.14 ± 2.55 | 4.49 ± 2.02 | 0.0034 |

total cladoceran density and 51.2%–80.3% of the biomass (Figs. 2C and 2D). Among the dominant cladocerans that comprised 1% or more of total cladoceran density at all sites, *Bosmina hagmanni*, *B. longirostris* and *B. deitersi* showed higher density and biomass in black water than in white water (Figs. 2C and 2D). In contrast, those of *Moina minuta* were higher in white water than in black water.

The density and biomass of copepods were also significantly higher in the black water river (763 ± 530 inds. m$^{-3}$; 3.37 ± 1.76 mg m$^{-3}$, mean of St. 4-5) than in the white water river (331 ± 164 inds. m$^{-3}$; 1.77 ± 1.22 mg m$^{-3}$, mean of St. 1-2) (Tables 3 and 4). In total, 25 species of copepods were observed (Table S2), among which (excluding copepodites) *Oithona amazonica* was the most dominant taxa in terms of density at all sites, contributing with 9.0%–40.6% to the total copepod density, while *Dactylodiaptomus pearsei* was the most important in terms of biomass (34.6–58.5%) (Figs. 2E and 2F). The highest density of *O. amazonica* was observed at the center (St. 4) of black water river (388 ± 566 inds. m$^{-3}$), followed by the confluence (349 ± 405 inds. m$^{-3}$).

Although there was no significant difference in the density of insect larvae between the white (3.8 ± 2.4 inds. m$^{-3}$, mean of St. 1-2) and black water rivers (5.0 ± 3.1 inds. m$^{-3}$, mean of St. 4-5), the biomass of insect larvae was significantly higher in the black water river (0.53 ± 0.39 mg m$^{-3}$) than in the white water river (0.22 ± 0.13 mg m$^{-3}$) (Tables 3 and 4). The density of insect larvae was highest in the bank (St. 5) of black water river, followed by the confluence (St. 3) and the bank (St. 1) of white water river (Fig. 2G). Chaoboridae (diptera) was numerically abundant in the black water river, while chironomidae (diptera) and coleoptera were dominant in the white water river (Fig. 2G, Table S3). The biomass of insect larvae was the highest in the bank of the black water river (St. 5) decreasing toward the bank (St. 1) of the white water river (Fig. 2H).

## Ordination of the mesozooplankton community

The MDS ordination plot and group-average clustering showed that mesozooplankton communities in the black water river were clearly separated from those in the white water river (Fig. 3). The result of ANOSIM test showed that the community structure between black and white water rivers was significantly different (Global $R = 0.622$, $P = 0.001$). The communities from the confluence were in between black and white water communities.

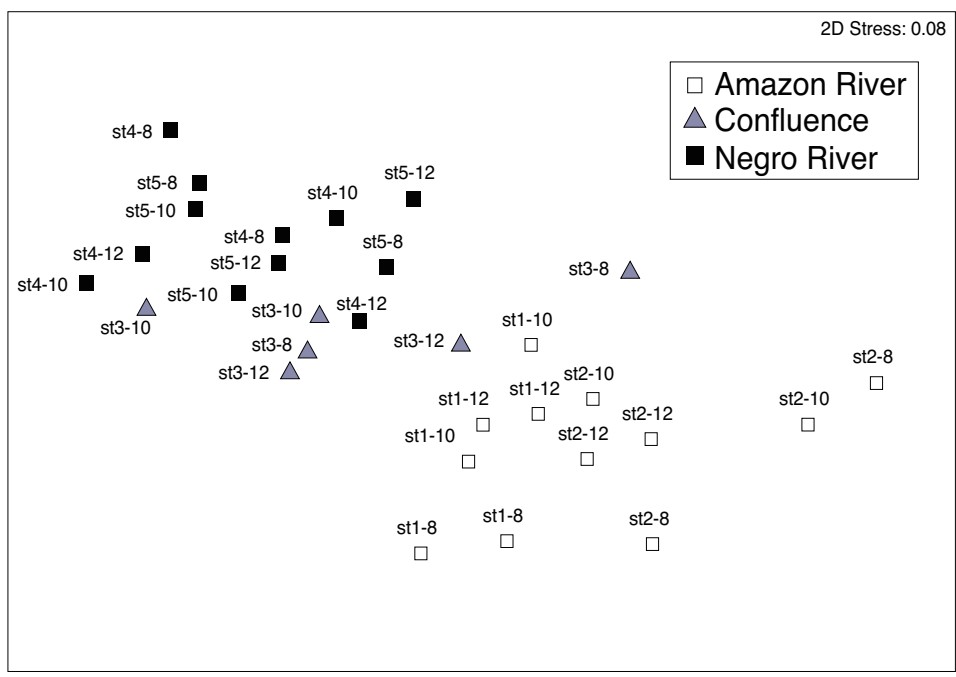

**Figure 3** **Non-metric multidimensional scaling (MDS) plots.** MDS plots showing similarity of meso-zooplankton community in different sites (the Amazon River; the Negro River; the confluence). Bray-Curtis similarities were calculated based on the square-root of abundance. The legends above each symbol indicate sampling station (st1-5) and date of sampling (8–12 March 2012).

## Ichthyoplankton density and composition

The density of larval fish (ichthyoplankton) was significantly higher in the white water river (3.2 ± 3.1 inds. m$^{-3}$, mean of St. 1-2) than in the black water river (1.2 ± 1.3 inds. m$^{-3}$, mean of St. 4-5) ($t$-test, $P = 0.045$). The density of larval fish in the confluence (St. 3) (9.7 ± 2.5 inds m$^{-3}$) was significantly and 2.1–8.8 times higher than in all the other four sites (Tukey-Kramer, $df = 29$, $P < 0.01$) (Fig. 4). Characiformes were the most dominant group in the confluence, contributing with 47.2% to the total larval fish density, followed by Pimelodidae (siluriformes, 34.5%). The larval fish density at the bank of white water river (St. 1) was the next abundant (4.6 ± 3.7 inds m$^{-3}$). Auchenipteridae (siluriformes) were only sampled at the banks of both white (St. 1) and black water rivers (St. 5), while clupeiformes were only observed in the center of the white water river (St. 2).

## DISCUSSION
### Water properties of the two rivers

This study describes the density and biomass of mesozooplankton and ichthyoplankton across the Negro River (black water) and the Amazon River (white water) in the center of the Amazon basin to elucidate the distributional differences between the two rivers and their confluence zone, which were not previously well-described quantitatively. The water properties of the two rivers were distinct: surface water temperatures and transparency were

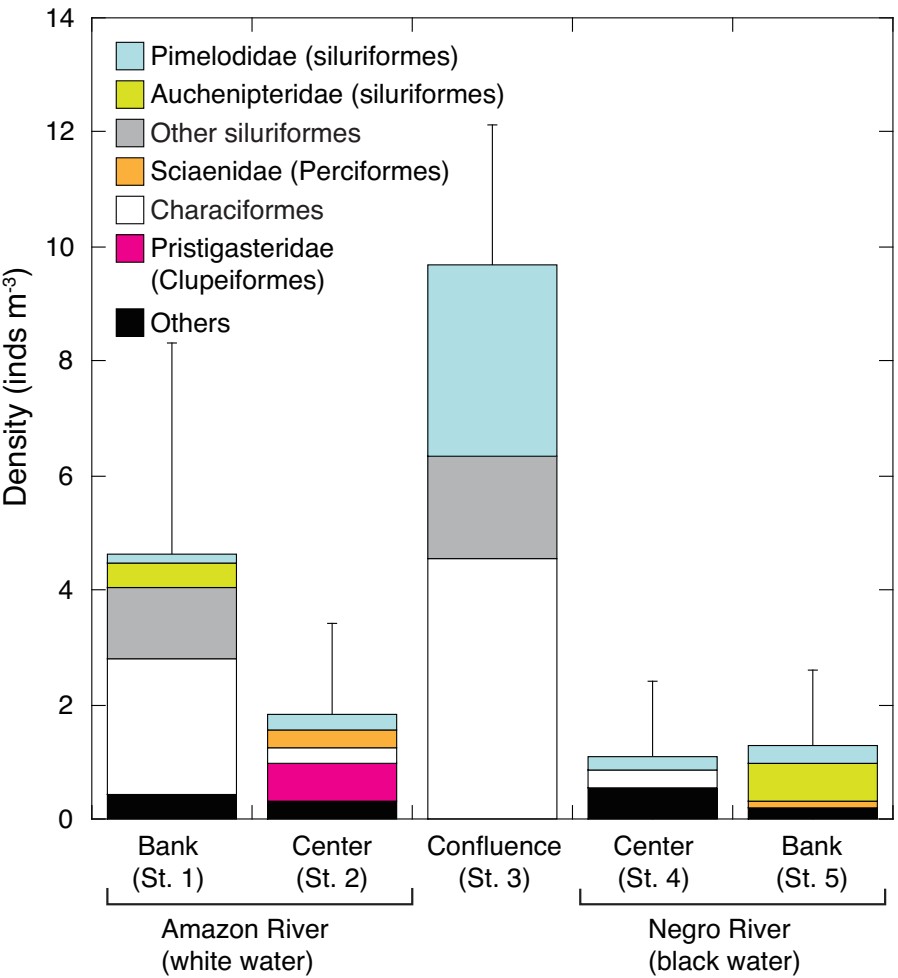

**Figure 4 Spatial variation in abundance of ichthyoplankton.** Average (mean ± SD) density of ichthyoplankton community in the surface water of the Amazon River (St. 1-2), the confluence (St. 3), and the Negro River (St. 4-5). Error bars represent standard deviation (SD) of ichthyoplankton abundance for six replicate measurements. Each legend category indicates the proportion of each taxon per mean.

higher in black water rivers, while chlorophyll and particulate organic matter concentrations were higher in white water rivers.

Surface water temperature in black water was higher by 0.6 °C on average than white water, which is congruent with previous studies reporting higher temperature by 1 °C in the Negro River (*Franzinelli, 2011*). The higher water temperature in the Negro River may result from its darker color and slower current speed compared to the Amazon River (0.1–0.3 m s$^{-1}$ vs. 1.0–1.3 m s$^{-1}$) (*Moreira-Turcq et al., 2003*; *Filizola et al., 2009*; *Franzinelli, 2011*).

The mean concentration of chl-*a* in the white water river (3.8 μg L$^{-1}$) was higher than that in the black water river (2.0 μg L$^{-1}$) in this study. Although concentrations are much different between lakes and rivers, a similar pattern was previously reported in floodplain lakes, where surface water chl-*a* concentration was higher in lakes associated with the Amazon River (white water, 50–80 μg l$^{-1}$) than in lakes adjacent to the Negro

River (black water, 10–20 µg l$^{-1}$) (*Fisher & Parsley, 1979*; *Trevisan & Forsberg, 2007*). Higher chl-*a* concentration in white water lakes is due to the higher concentrations of inorganic nutrients derived from the Amazon River (*Trevisan & Forsberg, 2007*). However, in the Amazon River, the production of phytoplankton is not likely because of poor light penetration due to high turbidity (euphotic depth: ca. 0.3 m), where the mixing depth was probably always down to the bottom due to turbulence associated with the strong current, making respiration higher than photosynthesis (*Fisher & Parsley, 1979*). Therefore, the higher chlorophyll concentration in the Amazon River probably results from the input of more productive environments such as the adjacent lakes (*Fisher & Parsley, 1979*).

## Mesozooplankton difference between black and white water rivers

As the MDS and ANOSIM analyses clearly indicated, the present study revealed that the compositions of mesozooplankton assemblages differ between the white water of the Amazon River and black water of the Negro River. We also found a higher density of mesozooplankton communities in black water river compared to white water river. The density of zooplankton in tropical large rivers depend largely on the supply from adjacent lentic sources (standing water bodies) connected to the river such as channel and floodplain habitats (*Rzoska, 1978*; *Saunders & Lewis, 1988a*; *Saunders & Lewis, 1989*; *Basu & Pick, 1996*; *Reckendorfer et al., 1999*; *Górski et al., 2013*). The zooplankton sampling period in this study corresponds to the rising water period (March), where rising riverine water starts to wash out ambient zooplankton from associated lentic sources into the rivers (*Saunders & Lewis, 1988a*; *Saunders & Lewis, 1988b*; *Saunders & Lewis, 1989*). Assuming that adjacent lentic areas (e.g., floodplain lakes) are a major source of zooplankton in river systems in this study, there may have been more zooplankton transport from stagnant water bodies connected to the Negro River (black water) compared to those of the Amazon River (white water). However, there are fewer lakes in the Negro River floodplain than in the floodplains of white water rivers because of the lower hydrodynamics (*Junk et al., 2015*). Previous studies from floodplain lakes in the center of the Amazon basin reported that the density of mesozooplankton (cladocerans and copepods) was 2–25 fold higher in black water lakes associated with the Negro River than in white water lakes during rising-high water periods (Feb-June) (*Brandorff, 1978*; *Hardy, 1980*), which might explain the higher mesozooplankton density in the black water river in this study.

Reproduction of zooplankton in the flowing waters can also increase density at a low flow rate (*Bertani, Ferrari & Rossetti, 2012*). River zooplankton are unable to reproduce in flow speed exceeding 0.4 m s$^{-1}$ (*Rzoska, 1978*) and thus lower residence time can mean a lower zooplankton density (*Basu & Pick, 1996*). Considering that the flow speed of the Amazon River (Rio Solimões) exceeds 1.0 m s$^{-1}$ (*Filizola et al., 2009*), reproduction of zooplankton may be impossible in this white water river. Large amounts of inorganic suspended particles in white water river may also negatively influence zooplankton density in this system (*McCabe & O'Brien, 1983*; *Kirk & Gilbert, 1990*; *Junk & Robertson, 1997*). Indeed, zooplankton density in the white water river was higher in the bank than at the center, suggesting that adjacent lentic sources are the primary source of zooplankton in this white river system. On the contrary, mesozooplankton in the Negro River showed higher

density in the center of the river than in the bank, implying that zooplankton reproduction occurs in this slower current of black water river (0.1–0.3 m s$^{-1}$) (*Moreira-Turcq et al., 2003*; *Franzinelli, 2011*). During the low-water periods, most floodplain lakes are isolated from the active river channel, which leads to a lower supply of zooplankton to the large rivers (*Saunders & Lewis, 1989*). The density of zooplankton in tropical large rivers during low-water period might be primarily determined by the reproduction of zooplankton in the rivers. In summary, the higher supply of zooplankton from adjacent lentic water bodies (such as floodplain lakes) and/or possible reproduction could help to explain why mesozooplankton density was higher in the black water river compared to the white water river at the rising water period.

## Mesozooplankton and ichthyoplankton in the confluence

As previously examined in oceanic frontal boundaries between river plumes and adjacent marine waters (*Morgan, De Robertis & Zabel, 2005*; *Walkusz et al., 2010*), convergent flow at the boundary between distinct water masses functions to concentrate planktonic organisms. However, an exceptionally high zooplankton number, as often seen in oceanic fronts (*Morgan, De Robertis & Zabel, 2005*), was not observed in the confluence boundary in this study. The highest average density of mesozooplankton was observed in the center of black water river (the Negro River), though zooplankton biomass was similar between the confluence and the black water river. Unlike oceanic fronts, where riverine freshwater plumes stand still facing the coastal marine water, which enhances the mechanical concentration of zooplankton (*Morgan, De Robertis & Zabel, 2005*), the black and white water rivers in the present study flow down together (but without mixing), probably making the zooplankton concentration less distinguished in the boundary zone. However it should be noted that the mesozooplankton density and biomass in the confluence was far higher than that in white water river, and the density and biomass of mesozooplankton in the confluence sometimes exceeded those in the Negro River. Zooplankton density can be higher due to accumulation at the areas where the velocity of the water current drops in the confluence of rivers (*Bolotov, Tsvetkov & Krylov, 2012*). The different current speeds between the Amazon River ($\sim$0.3 m s$^{-1}$) and the Negro River (1.0$\sim$m s$^{-1}$) (*Moreira-Turcq et al., 2003*) could be a trigger for the episodic higher density of mesozooplankton in the confluence.

The plankton net used in this study was not strictly designed for collection of ichthyoplankton (usually a net with a larger mouth and mesh opening is used), thus our net may have misrepresented the number and species richness of fish larvae. Yet we found significantly higher density of fish larvae in the confluence throughout the study period, supporting the hypothesis that the confluence between white and black water rivers functions as an ecological concentrator of ichthyoplankton (*Morgan, De Robertis & Zabel, 2005*).

Then the question arises as to why ichthyoplankton density was high in the confluence boundary zone. Previous studies revealed that turbidity affects predation risk through less predation risk in turbid water because turbidity reduces the distance at which predator–prey interactions occur (*Abrahams & Kattenfeld, 1997*; *Ranåker et al., 2014*). Turbulence is also one of the factors affecting predation risk through lower risk in more turbulent habitats

(*Weissburg & Zimmer-Faust, 1993*). In black water rivers, potentially higher predation risks for larval fish would be expected given that larvae can be more easily seen by predators due to fewer suspended solids (*De Lima & Araujo-Lima, 2004*). On the contrary, white waters with high suspended solids are considered to be safer places for larval fish because of lower transparency and higher turbulence, which may act as refuge from predators (*Weissburg & Zimmer-Faust, 1993*; *De Lima & Araujo-Lima, 2004*). Therefore the confluence zone can be a boundary interface between high and low predation pressures for fish larvae. From the perspective of food availability (at least for zooplanktivorous fish), the confluence between white and black waters is sandwiched by both environments with low and high food concentrations. Fish larvae may find more prey in black water river, yet fish larvae density was the lowest in the Negro River, suggesting higher predation pressure in black water river even in a food-rich environment. Therefore, the confluence zone between black and white water rivers may function as a boundary layer that has benefits from both low predation risk and high food concentrations for fish larvae. Similar observation was previously reported on the surface of a freshwater marsh where a great density of small fish was found at a site adjacent to the shallow depositional bank compared to a site adjacent to the deeper erosional bank, due to greater prey availability and less predator pressure at the site adjacent to the depositional habitat (*McIvor & Odum, 1988*). In summary, the combined effects of food availability and predator avoidance form a plausible explanation for the high density of ichthyoplankton in the confluence zone of black and white water rivers. The lower C/N ratio of POM found in the confluence compared to the adjacent rivers may be the result of higher heterotrophic activity in this boundary zone since the C/N ratio of carnivorous fish feces is generally very low (*Smriga, Sandin & Azam, 2010*).

## CONCLUSION

We found that mesozooplankton density and biomass were higher in the black water of the Negro River compared to the muddy white water of the Amazon River, probably due to a higher supply of zooplankton from lentic waters adjacent to the Negro River and/or reproduction. An exceptionally high mesozooplankton density was not observed in the confluence boundary between the two rivers; nonetheless, we found that the confluence zone acts as an aggregator of ichthyoplankton. The confluence boundary between black and white water rivers may function as a boundary layer that offers benefits of both high food (zooplankton) concentrations from black water river and low predation risk from white water river. This forms a plausible explanation for the high density of ichthyoplankton in the confluence zone.

## ACKNOWLEDGEMENTS

The authors thank F Mariano, J Pablo TCA, VH Estefes and ES Hase for field assistance; DR Freitas (Centro de Biotecnologia da Amazonia) for help in chlorophyll measurements; and three anonymous reviewers for helpful comments on this manuscript. The first author thanks S Sato and T Toda (Soka University) for help in visa application and providing a

plankton net; and J. Hashimoto (Nagaseya, an aquarium shop in Tokyo) for providing a motivation to conduct this research.

### Funding
This study was supported by the JSPS Fellowship for Study Abroad. There was no additional external funding received for this study. The funders had no role in study design, data collection and analysis, decision to publish, or preparation of the manuscript.

### Grant Disclosures
The following grant information was disclosed by the authors:
JSPS Fellowship for Study Abroad.

### Competing Interests
The authors declare there are no competing interests.

### Author Contributions

- Ryota Nakajima conceived and designed the experiments, performed the experiments, analyzed the data, contributed reagents/materials/analysis tools, wrote the paper, prepared figures and/or tables, reviewed drafts of the paper.
- Elvis V. Rimachi performed the experiments, analyzed the data, wrote the paper, reviewed drafts of the paper.
- Edinaldo N. Santos-Silva analyzed the data, contributed reagents/materials/analysis tools, wrote the paper, reviewed drafts of the paper.
- Laura S.F. Calixto and Rosseval G. Leite analyzed the data, reviewed drafts of the paper.
- Adi Khen wrote the paper, prepared figures and/or tables, reviewed drafts of the paper.
- Tetsuo Yamane contributed reagents/materials/analysis tools, wrote the paper, reviewed drafts of the paper.
- Anthony I. Mazeroll wrote the paper, reviewed drafts of the paper.
- Jomber C. Inuma and Erika Y.K. Utumi performed the experiments, reviewed drafts of the paper.
- Akira Tanaka conceived and designed the experiments, performed the experiments, contributed reagents/materials/analysis tools, wrote the paper, reviewed drafts of the paper.

### Data Availability
The raw data has been supplied as a Supplementary File.

### Supplemental Information
Supplemental information for this article can be found online at http://dx.doi.org/10.7717/peerj.3308#supplemental-information.

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
