# Peer review of "The density and biomass of mesozooplankton and ichthyoplankton in the Negro and the Amazon Rivers during the rainy season: the ecological importance of the confluence boundary"

_PeerJ, doi:10.7717/peerj.3308_

## Round 0.1 · original submission · Major Revisions

Your manuscript has to be improved according to the suggestions given by the reviewers. Special focus should be considered on the validity of the findings and improving of the discussion section.

Reviewer 1 ·

Basic reporting

The English is clear. The background and literature are mostly sufficient (but see comments below). The structure is fine. Figures and tables are good. Raw data are complete.

The authors explain the background of the boundary zones clearly and develop a hypothesis and two research questions from it. However, I think that the authors could clarify the background on the fish communities. For example, at what developmental stage are the fish in these communities planktivores? The authors mention juveniles in the introduction, but sampled ichthyoplankton, which I think of as larval fish. Also, on lines 72-75 and lines 376-378--are the fishermen catching juveniles or adults? Either way, I think that this anecdotal evidence is unnecessary.

I think a paragraph on the structure of the food web could help. Given that the ichthyoplankton may not have been sampled thoroughly (lines 114-116), focusing the manuscript on mesozooplankton might be better. Some of the wording in the introduction may need to be changed to clarify that this study measured potential prey concentrations, but not juvenile fish or upper trophic levels.

Experimental design

Research within scope of PeerJ. Questions and knowledge gap are well-defined. Methods and techniques described well, but see questions below.


The mesozooplankton sampling method followed standard methods (eg, transect sites, mesh size, sample analyses/identification), but the ichthyoplankton sampling did not (as noted by the authors, lines 114-116).

Since the river levels and environmental conditions change over the year (lines 173-187), why was the plankton sampling conducted only during March? When are juvenile fish in the system? How does the ecosystem change during other times of the year (the discussion on line 293 begins to explain this)? Does this limit how much can be inferred from the sampling data?

Validity of the findings

In general, the conclusions are supported by the data, although I think that 1) clarification is needed about the juvenile fish and ichthyoplankton part of this study and 2) discussion of how representative March sampling is for the rest of the year should be discussed. The new hypothesis that the black water/white water boundary may offer the optimal location for high foraging opportunity and reduced predation risk is interesting, although more citations (or data) on predation risk would be useful here.

Additional comments

I think that the discussion could be restructured so that the discussion of the mesozooplankton at the confluence (line 327) comes first, since that is what the manuscript’s hypothesis was about.

Reviewer 2 ·

Basic reporting

Overall, the ms is rather well written in an acceptable, professional and clear English Language.
Literature references and background knowledge are documented well. On the contrary, I often found very generous background citation, sometimes too many for a single statement.
The article structure is well defined, even if it has to be improved to make the article acceptable for the publication.
Graphical and table representation is very carefully done, there are just some minor errors and adjustments to be done.
The outcomes are relevant and the hypothesis is properly tested. The work is self-contained, but possibly lacking of wider view and ecological significance, out of the study area.

Experimental design

Experimental design fulfills all necessary criteria.

Validity of the findings

The findings are discussed well, supportd by robust data analysis
According to my opinion, this work just lacks wider discussion, encouriging even speculation or the development of different possible scenarios.

Additional comments

I'm sending as pdf file in attachment detailed list of mainly minor issues that need to be faced by the authors.
I consider that the ms could be worth of publishing for its potential quality and content, but cannot be published in the present form. Because of its potential to be strongly improved, with lots of minor issues that should be reviseted, I would recommend major revision.

Overall, the graphical presentation and description of the outcomes of the research are satisfying. The statistical analysis is well done and robust.
The parts of the ms that have to be mainly revisited are the Introduction and Methodology section. References must be carefully checked and FORMATTED throughout the text!!!
Discussion and conclusion are lacking major ecological implications, based on the present data and comparisons with other similar situations if possible.

Annotated reviews are not available for download in order to protect the identity of reviewers who chose to remain anonymous.

---

## Round 0.2 · Major Revisions

Your paper still needs to be improved. Please follow all suggestions given with special interest those related to discussion try to correlate the environmental factor with the biological data..

Reviewer 1 ·

Basic reporting

NC

Experimental design

NC

Validity of the findings

NC

Additional comments

The authors have answered my previous comments well and revised their manuscript. I think that the manuscript meets all of the requirements for publication.

Reviewer 3 ·

Basic reporting

This study has a great potential to be published in this journal, but not in the current form. Although background was properly provided, discussion seems less scientific with few discussion on worldwide comparisons to support conclusions. Figures were well presented and easy to follow along the text, however this is not strong related to the sampling design provided in methods (see comments below). English is suitable.

Experimental design

My first concern is that authors tried to hide that this is a short-term study. Sampling design and hypothesis are not straightforward related. This hypothesis must be reformulated in order to assume that this is a short-term study. Sampling design was performed to induce results using replicates of day and night, bottom and surface, but theses assumptions were not considered within the hypothesis. For more comments, see below. My suggestion is to change the sampling design, avoiding day and night approaches, or provide a more robust data by assuming these factors.

Validity of the findings

If the experimental design is provided in the current form, the results are inconsistent and not related to the hypothesis (which is wrong). In addition, I suggest to correlate environmental factor with biological data to corroborate all results and avoid speculative conclusions (seen bellow).

Additional comments

This study has a great potential to be published in this journal, but not in the current form. Although background was properly provided, discussion seems less scientific with few discussion on worldwide comparisons to support conclusions. Figures were well presented and easy to follow along the text, however this is not strong related to the sampling design provided in methods (see comments below). English is suitable.
My first concern is that authors tried to hide that this is a short-term study. Sampling design and hypothesis are not straightforward related. This hypothesis must be reformulated in order to assume that this is a short-term study. Sampling design was performed to induce results using replicates of day and night, bottom and surface, but theses assumptions were not considered within the hypothesis. For more comments, see below. My suggestion is to change the sampling design, avoiding day and night approaches, or provide a more robust data by assuming these factors.
If the experimental design is provided in the current form, the results are inconsistent and not related to the hypothesis (which is wrong). In addition, I suggest to correlate environmental factor with biological data to corroborate all results and avoid speculative conclusions (seen bellow).


ABSTRACT
 It would be interesting if values of abundance and biomass were provided in the abstract, in parenthesis.
 In my conception, abundance in ecology is a term used to define the amount of a resource, and this resource can be represented in terms of density or biomass (both are abundances). See if it is more plausible to change “abundance” for “density” here and in the entire text.
Lines 26-29: This hypothesis must be reformulated in order to assume that this is a short-term study. In addition, why there is no attempt to use day and night factors for testing such hypothesis? This is what the sampling design was proposed for.

Lines 29-31: “Our results show that mesozooplankton abundance and biomass were higher in the black-water river compared to the white-water river…”
OK, what is the P-value for this statistical difference?

Lines 31-32: “…however an exceptionally high mesozooplankton abundance was not observed in the confluence boundary.”
However, in figure 3 it seems that there is no difference between confluence and black waters (error bars overlap). What do you mean? It must be clear.

Lines 32-33: “Nonetheless we found the highest abundance of ichthyoplankton in the confluence boundary, being up to 9-fold higher than in adjacent rivers.”
The density of ichthyoplankton is higher in the confluence, but not statistically different from that of the bank of the Amazon River. In such case, it is wold be more plausible to change this sentence for “…being up to 9-fold higher than in adjacent rivers, except than in the bank region of the Amazon River.” What is the P-value.

Lines 34-36: “The confluence boundary between black and white water rivers may function as a boundary layer that offers benefits of both high zooplankton prey concentrations (black-water) and low predation risk (white-water).”
This sentence is very confusing. In lines 31-32, authors asserted that the abundance of zooplankton was not too high in the confluence zone. But now it is. In addition, what is the explanation for low predation risk in white-water?
I think it is very confusing. I this study black waters are clean, while white waters are darker. Se if would be more interesting to change these nomenclatures throughout the text.

INTRODUCTION
Suitable. Although, hypothesis is not well proposed according to methods.
MATERIAL AND METHODS
 Why did you chose vertical tows, if there is no attempt to study surface and bottom trophic structures?
 Assert that the experimental design used as replicates 3 sampling during the day and during the night for each site.
Lines 146-144: Remove this sentence. The study regards mesozooplankton. This is already clear.
Lines 154-155: This was not considered in the hypothesis. Also, bar graphs (fig. 3 and 5) did not consider average densities and biomass during day and night, such as was proposed in the sampling design. Why not? This is very confusing and inconsistent.
 There is no table with statistical results. It must be provided
 There is no analyses to identify correlations between mesozooplankton and ichthyoplankton with environmental factors (Chl-a, POC, PON, temperature, secchi depth). This would result in better conclusions for a short-term study.
RESULTS
 In the hydrological data section, I suggest to change the subheading for “Environmental factors and the structure of the confluence zone”. This section must explain the environmental variability and the boundary limits to introduce your following results, but not to corroborate results of a short-term experiment. Try to focus on the differences of water masses. In addition, I suggest to separate the nutrients and Chl-a parameters in another section.
Lines 179-183: This can explain why there is a higher abundance of mesozooplankton in black water. However, no correlation analyses was provided.
Lines 191-194: I think there are some homogeneous groups and these are known as exceptions. “The mesozooplankton abundance and biomass in the center of black water river significantly exceeded those of white water river.” However, for the biomass, it was observed an exception, since there is no difference between black water and the bank of the Amazon River.

Lines 235-237: Except when compared to the bank of the Amazon River.

Figure 3 and 5: Authors must use “average abundance and biomass”. Also explain what is the meaning of the each colour. Are they the proportions of each taxon per mean? Also, bar graphs did not consider average densities and biomass during day and night, such as was proposed in the sampling design. Why not? This is very confusing and inconsistent.

DISCUSSION
Discussion is very regional. A greater search must be done, to increase the reader’s interest (world-wide comparisons). Focus on short term studies. These are important to closely relate sharp/punctual moments of patterns of use of resources.
Lines 320-323: This sentence seems speculative, since there is no analysis to corroborate this conclusion.

---

## Round 0.3 · accepted · Accept

Thank you very much for improving your manuscript which now is going to be published.